# Oncogenic Role of Connective Tissue Growth Factor Is Associated with Canonical TGF-β Cascade in Colorectal Cancer

**DOI:** 10.3390/genes13040689

**Published:** 2022-04-14

**Authors:** Shaghayegh Hosseini, Leili Rejali, Zahra Pezeshkian, Mahtash Malekian, Nayeralsadat Fatemi, Noshad Peyravian, Mahrooyeh Hadizadeh, Zhaleh Mohsenifar, Binazir Khanabadi, Maral Farzam, Ghazal Sherkat, Hamid Asadzadeh Aghdaei, Ehsan Nazemalhosseini Mojarad, Maziar Ashrafian Bonab

**Affiliations:** 1Basic and Molecular Epidemiology of Gastrointestinal Disorders Research Center, Research Institute for Gastroenterology and Liver Diseases, Shahid Beheshti University of Medical Sciences, Tehran 19857-17411, Iran; shaghayeghhosseiny1990@gmail.com (S.H.); leilirejali@gmail.com (L.R.); zahrapezeshkian@yahoo.com (Z.P.); mahtash.malekian@yahoo.com (M.M.); n_fatemi_1363@yahoo.com (N.F.); n.peyravian@gmail.com (N.P.); binazir.khanabadi74@gmail.com (B.K.); hamid.assadzadeh@gmail.com (H.A.A.); 2Faculty of Health and Wellbeing, Canterbury Christ Church University, Canterbury CT1 1QU, UK; m.hadizadeh1176@canterbury.ac.uk; 3Department of Pathology, Taleghani Hospital, Shahid Beheshti University of Medical Sciences, Tehran 19839-63113, Iran; mohsenifar@sbmu.ac.ir; 4Gastroenterology and Liver Diseases Research Center, Research Institute for Gastroenterology and Liver Diseases, Shahid Beheshti University of Medical Sciences, Tehran 19857-17411, Iran; marall_jeiilan@hotmail.com; 5Medicin Faculty of Mashhad Branch, Islamic Azad University, Mashhad 91871-47578, Iran; gsherkat@yahoo.com; 6Kent and Medway Medical School, Canterbury CT2 7FS, UK

**Keywords:** MMP-1, CTGF, canonical TGF-β signalling, CRC

## Abstract

TGF-β signaling pathways promote tumour development and control several downstream genes such as CTGF and MMPs. This study aimed to investigate the association between CTGF and MMP-1 mRNA expressions with clinicopathological status and survival rate in colorectal cancer patients. We investigated expression levels of CTGF and MMP-1 genes in paraffin-embedded tumours and adjacent normal tissue blocks (ADJ) by Real Time-PCR. Then, the expression of Smad2 and Smad4 proteins in the TGF-β canonical pathway was evaluated by immunohistochemistry. Finally, the correlation between CTGF, MMP-1, and the canonical TGF-β-signalling pathway with the clinicopathological features was investigated. Expression levels of MMP-1and CTGF were higher in tumours compared with adjacent normal tissues. Overexpression levels of MMP-1 and CTGF were associated with lymph node metastasis, distant metastasis, tumour histopathological grading, advanced stage, and poor survival (*p* < 0.05). Additionally, a significant association between the upregulation of MMP-1 and tumour location was noted. Upregulation of Smad2 and Smad4 proteins were also significantly correlated with lymph node metastasis, distant metastasis, advanced stage, and poor survival (*p* < 0.0001). This study showed that canonical TGF-β signalling regulates both CTGF and MMP-1 expression and CRC progression. Moreover, TGF-β signalling and its downstream genes could be used as novel biomarkers and novel approaches for targeted therapy in CRC.

## 1. Introduction

Colorectal cancer (CRC) is the third leading cause of cancer mortality and the fourth most frequently occurring cancer in the world [1,2]. Many signalling pathways are involved in CRC initiation and progression, one of which is transforming growth factor-β (TGF-β). It has been shown previously that TGF-β signalling plays a crucial role in cancer deployment via both Smad-dependent and Smad-independent pathways [3,4,5,6]. At the early stages of CRC, TGF-β inhibits tumorigenesis by stimulating apoptosis in cancerous cells. However, in the advanced stages of CRC the upregulation of TGF-β induces metastasis and malignancy features. Previous studies have demonstrated the TGF-β pathway affects CRC pathogenesis through upregulation of CTGF via the canonical Smad-dependent pathway [3,7,8,9]. CTGF is a transcriptional target of TGF-β signalling and induces angiogenesis and invasion by the regulation of matrix metalloproteinases (MMPs) [10,11], which are extracellular matrix (ECM) degradation enzymes and play key roles in tumour progression through digestion of ECM and providing a suitable condition for tumour angiogenesis, invasion, and metastasis [12,13]. Overexpression levels of MMP-1 have been reported in various cancer tissues, where MMP-1 overexpression was associated with angiogenesis, lymph node metastasis, and poor prognosis in CRC [14,15,16,17,18].

The discovery of potential new biomarkers for CRC is essential for improvements in early diagnosis and improvements in patient outcomes. This study aimed to investigate the association between the TGF-β/Smad signalling pathways and CTGF and MMP-1 expression, with clinical status and survival rates in CRC patients.

## 2. Materials and Methods

### 2.1. Patients and Sample Collection

In this prospective study, we utilized 122 Formalin-Fixed Paraffin-Embedded (FFPE) tissue blocks of surgical samples from CRC patients who were referred to Taleghani and Shohada Hospitals, Tehran, Iran, from 2015 to 2018. Additionally, 20 FFPE samples with normal pathologic reports assigned were added as the control group. Tumour and ADJ normal resected sections are all confirmed by two expert pathologists. The inclusion criteria were considered as: (I) All entered CRC patients underwent colorectal surgery at the Taleghani or Shohada Hospitals; (II) All participants were followed up for 3 years and their clinicopathological reports were available; (III) None of those patients included received chemotherapy or radiotherapy before surgery. The exclusion criteria were determined as: (I) A history of preoperative radio-chemotherapy before surgery; (II) A history of familial polyposis and hereditary nonpolyposis CRC syndrome; (III) No valid clinical and pathological information was available, or the specimen quality or quantity was not acceptable.

The study was approved by the local research ethics committee of the Research Centre for Gastroenterology and Liver Diseases of Shahid Beheshti, University of Medical Sciences, Tehran (No. 136-1395). The Declaration of Helsinki as a statement of ethical principles for medical research was considered. Written informed consent was obtained from all patients and donors included in the study.

### 2.2. Clinicopathological Data

The present study consisted of 77 (63.1%) men and 45 (36.9%) women, with an age cut-off at 60 years, where 45% were designated under 60 and near 55% over. All patient’s demographic and clinicopathological parameters are presented in detail in Table 1. All clinical and histopathological reports were assigned by two expert pathologists.

### 2.3. RNA Isolation and cDNA Synthesis

Then, 12 µm thick sections were prepared from both tumour and normal paraffin-embedded tissue blocks. Sections were deparaffinized in 180 µL xylene and then rehydrated by twice incubating in 850 µL absolute ethanol for 3 min followed by PBS washing. Total RNA was extracted from each air-dried deparaffinized section of the tumour and adjacent normal tissues, 10 cm away from the canonical tumour region via RNeasy FFPE kit (Qiagen, Hilden, Germany) as per the manufacturer’s protocol and was stored at −80 °C until use. The yield of extracted RNA and the A_260_/A_280_ purity ratios were measured using Nanodrop (ND-1000 spectrophotometer-Thermo Fisher, Waltham, MA, USA). RNA integrity was determined by electrophoresis on a 1% agarose gel. QIAxcel capillary electrophoresis with a high-resolution cartridge, 25–500 nucleotide molecular marker, and 15–156 nucleotide align marker for detachment of segments generated by PCR was used for RNA quality measurement with new RNA Integrity Score. cDNA was synthesized using the PrimeScript 1st strand cDNA Synthesis Kit (Takara, Dalian, China) as per the manufacturer’s instructions.

### 2.4. Real Time PCR Analysis

For gene expression quantification, real-time-PCR was performed in an Applied Biosystems 7500 Real-Time PCR System using an SYBR Green Real-Time PCR Master Mix (Takara, Dalian, China). Specific primers are listed in Table 2. Thermal cycling conditions were as follows: 30 s at 95 °C, 95 °C for 5 s, 58 °C for 34 s, and a primer extension 60 °C for 34 s and 15 s at 95 °C, for 40 cycles. Relative expression of the selected genes was measured by normalizing to β-globin via the 2^−ΔΔCt^ method.

### 2.5. Immunohistochemistry and Evaluation of Staining

Immunohistochemistry (IHC) was used to evaluate the canonical TGF-β signalling pathway proteins including TGF-β, Smad2, and Smad4. In brief, 4-micron thick sections of paraffin-embedded tumoral tissues were laid on glass slides coated with Poly-L-lysine and incubated at 60 °C for 15 min; then the sections were deparaffinized and dehydrated by xylene and ethanol, and retrieval antigen was performed in a microwave (900 W, 27 min) by adding HCL to change pH to the extreme alkaline (pH = 9) and coolness. Hydrogen peroxidase 10% was used to block endogenous peroxidase activity and, thereafter, was rinsed in deionized water. The samples were blocked using blocking serum. Next, the prepared slides were incubated with the Smad2 (phospho S467, Abcam, Cambridge, UK), Smad4 (Anti Smad4-Phospho-T277Abgent), and anti-TGF-β (antibody-ab92486) antibodies for 30 min at 37 °C. The Mouse/Rabbit PolyVue Plus HRP/DAB Detection System (Diagnostic BioSystems, Pleasanton, CA, USA) was used for primary antibodies visualization according to the manufacturer’s instructions.

The slides were examined under a light microscope (Nikon, Tokyo, Japan) by two pathologists who were unaware of the clinical and genetic data. Mean values were approximated through the scanning of the entire tissue parts of all samples via two graded scales: negative, <10%, and positive, >10%. The positive controls were the following: (a) a healthy colonic tissue was taken as an internal control for β-globin IHC, and (b) a histologically diagnosed part of colon cancer tissues for nuclear positivity by β-globin IHC. Negative controls were obtained by omitting the primary antibody.

The immunoreactivity of each slide was scored based on the Allred scoring system. Expression of TGF-β in tumour stroma and simultaneous expression of pSmad2 and Smad4 in the nucleus were assessed. Tumours were thus divided into two groups, including TGF-β positive, in which all the three genes were upregulated, and TGF-β negative, in which all the genes or at least two were downregulated.

### 2.6. Statistical Analysis

All gene expression experiments were performed in triplicate and repeated three times. The Relative Expression Software Tool (REST) was applied for the calculation of fold changes in gene expression. Statistical Package for Social Sciences SPSS (version 21) (SPSS Inc., Chicago, IL, USA) and Graph Pad Prism 8.0 software were used for analysing released data. The efficiency of gene expressions was assessed by the linear regression method. The Pearson, Chi-squared test (χ^2^), and Fisher’s exact test were used to determine the correlations between expressions and clinicopathological parameters in colorectal cancer patients. Non-parametric analysis (Mann–Whitney u-test) was used to compare gene expression in two different categories and one-way ANOVA (Kruskal–Wallis tests) was compared among more than two series. Survival analysis graphs were drawn by the Kaplan–Meier method. The *p*-value below 0.05 (*p* < 0.05) was considered statistically significant.

## 3. Results

### 3.1. High Expression of CTGF Contributes to Poor Prognosis in Colorectal Cancer Patients

CTGF mRNA expression in tumour tissues was found to be significantly higher in CRC tissues in comparison with adjacent normal tissues (*p* < 0.0001, Figure 1A). CTGF expression was significantly correlated with tumour location (*p* < 0.003), tumour differentiation (*p* < 0.0002), lymph node metastasis (*p* < 0.005), distant metastasis (*p* < 0.044), and tumour stage (*p* < 0.0001). However, no significant association was found between the expression level of CTGF and tumour size. Additionally, significant upregulation of CTGF was found in stage IV of disease and CRC samples with positive lymph node metastasis (Figure 2, Table 3).

Kaplan–Meier analysis for plotting survival curve data was used for CRC patients’ survival rate, these data documented that high CTGF expression was significantly related to poor survival (*p* < 0.01) (Figure 3).

### 3.2. MMP-1 Is Overexpressed in Colorectal Carcinoma Samples and Is Related to Poor Prognosis in CRC Patients

The mRNA expression level of MMP-1 was significantly higher in tumour tissues compared with adjacent normal tissues (*p* < 0.0253, Figure 1B). Our data demonstrated a significant association between MMP-1 mRNA level and clinicopathological parameters, including tumour size (*p* < 0.0153), cellular differentiation (*p* < 0.0225), tumour stage (*p* < 0.0016), lymph node metastasis (*p* < 0.0285), and distant metastasis (*p* < 0.0128, Figure 2). However, no significant relationship was observed between MMP-1 expression and tumour location (*p* < 0.593). The MMP-1 expression was significantly higher in CRC cases at stage IV and the up-regulation of MMP-1 was found in cases with lymph node metastasis as well as distant metastasis. The Kaplan–Meier analysis graph was plotted to correlate patient survival through the expression level of MMP-1. The overexpression of MMP-1 was significantly linked to a lower survival rate compared with cases with downregulation of MMP-1 (*p* < 0.05) (Figure 3).

### 3.3. CTGF Increases MMP-1 Expression in Colorectal Carcinomas

Spearman test for determining the correlation between designated genes was applied. A significant correlation was observed between the expression of levels MMP-1 and CTGF in CRC patients (Spearman’s rank test, r = 0.4224, *p* < 0.0001, Figure 3).

### 3.4. TGF-β Signalling Pathway Has a Multifaceted Role in Colorectal Cancer

We analysed and scored expression levels of TGF-β, Smad2, and Smad4 proteins in 122 FFPE specimens by IHC staining and investigated the association between protein expression and the clinicopathologic characteristics of CRC patients.

Based on our data, 66.40% (81 cases) of tumours had high expression of Smad2 followed by positive TGF-β signalling (described above and 33.60% (41 cases) of tumours represented a low expression of Smad2 with negative TGF-β signalling through cytoplasmic staining. All positive samples had a homogeneous pattern in staining. Figure 4A,B illustrates the low and high expression of Smad2 in the cytoplasm.

Moreover, 82.52% (104 cases) of tumours with positive TGF-β signalling showed a high expression of Smad4 in the nucleus and 17.48 % (18 cases) of tumours indicated low expression of Smad4 with negative TGF-β signalling. All positive samples had a homogeneous pattern of staining. Figure 4C,D shows differences in Smad4 expression levels between low and high groups.

TGF-β as a nucleocytoplasmic shuttle was expressed in the cytoplasm and nucleus. Figure 4E,F depicts images of TGF-β protein expression 72.13% (88 cases) of participating patients illustrated high expression of TGF-β protein in the tumour microenvironment, while the rest 27.87% (34 cases) are categorized in low TGF-β expression category.

The obtained data from the current study revealed a significant correlation between positive TGF-β signalling and lymph node metastasis, distant metastasis, tumour stage, and overall survival (*p* < 0.001). However, no significant association was seen between age, gender, tumour location (*p* = 0.826), tumour size (*p* = 0.653), and tumour differentiation (*p* = 0.847).

### 3.5. The Effect of CTGF and MMP-1 mRNA Expression on TGF-β Protein Expression

As shown in Figure 4, mRNA expression levels of CTGF and MMP-1 were determined in groups with increased TGF-β protein and decreased TGF-β protein levels. The gene expressions levels of CTGF and MMP-1 were significantly increased (*p* = 0.0002, *p* = 0.0356, respectively) in tumours with a high level of TGF-β protein (Figure 4).

## 4. Discussion

The detection of CRC progression using biomarkers and novel approaches for early diagnosis/improved treatment is essential to reducing mortality rates in CRC [19]. In this study, we investigated the correlation of the canonical TGF-β signalling pathway with its downstream genes, including CTGF and MMP-1, and their effect on clinicopathological features in CRC patients. TGF- β signalling is known as a critical signalling pathway that has a tumour-suppressive role in the early stages of cancer and pro-metastatic function in advanced stages [20,21,22,23]. The Smad proteins, such as Smad2 and Smad4, are master regulators for the canonical TGF-β signalling [24,25,26]. Furthermore, the TGF-β signalling pathway regulates many downstream target genes, which are involved in cell growth, differentiation, apoptosis, and invasion [27]. TGF-β signalling activation in tumour microenvironments induces CMS4 mesenchymal phenotype (Malignant CRC phenotype), which is linked to worsening outcomes, via differentiation of mesenchymal stem cells to cancer-associated fibroblast, induction of JAK/STAT pathway, and suppression of tumour immunity [28,29]. Moreover, there are many challenges in cancer treatment using TGF-β signalling inhibition because of some side effects on patients. Hence, downstream pathways of TGF-β signalling could be utilized for targeted therapy with fewer side effects [30,31]. In the current investigation, obtained results confirmed previous findings of the association of activated TGF- β signalling with lymph nodes metastasis, distant metastasis, advanced stages, and poor survival [32,33,34]. CTGF gene is a downstream target of TGF-β signalling in the canonical pathway and a regulator of cell proliferation in pathological conditions [35,36]. The upregulation of CTGF leads cancerous cells to angiogenesis and metastasis, also the high expression level of CTGF has been detected in CMS4 colorectal cancer [10,37,38]. In this study, we found significant correlations between the upregulation of CTGF, tumour location, cellular differentiation, lymph nodes metastasis, distant metastasis, advanced stages, and poor survival, also upregulation of CTGF at stage IV was associated with lymph nodes metastasis; these results agree with previous studies [10,39]. Indicating that TGF-β signalling is an important regulator for CTGF expression and CRC development.

The MMP-1 protein is one of the most important matrix metalloproteinases, and a member of the collagenases, with a key role in CRC initiation and development [13,40]. In addition, CTGF can stimulate angiogenesis in cancerous cells via the regulation of MMPs [41,42,43,44]. In the current investigation, our data illustrated the overexpression of MMP-1 in CRC specimens, which confirmed the oncogenic role of MMP-1 in colorectal cancers [45]. Additionally, we observed that upregulation of MMP-1 was associated with tumour size, cellular differentiation, lymph nodes metastasis, distant metastasis, and poor survival. These results are again in agreement with previous findings [14,40,46,47,48]. It is worth mentioning that we found a significant correlation in MMP-1 overexpression among stage IV, lymph nodes metastasis, and distant metastasis. Therefore, it seems that MMP-1 plays a critical role in metastasis in advanced stages of CRC. It is interesting to note that, CTGF and MMP-1 presented a significant positive association with each other also. This result was reported for the first time in the Iranian population. Additionally, statistical analysis showed that the expression levels of CTGF and MMP-1 have been affected by TGF-β signalling. Based on the data presented here, CTGF and MMP-1 are regulated by TGF-β signalling, and TGF-β plays a crucial function through angiogenesis and malignancy in CRC.

## 5. Conclusions

In conclusion, this study showed that canonical TGF-β signalling is a master regulator for MMP-1 and CTGF and can control malignancy features in CRC. Moreover, downstream genes of TGF-β signalling, such as MMP-1 and CTGF, could be used as biomarkers for CRC detection to improve early diagnosis and patient outcomes.

## Figures and Tables

**Figure 1 genes-13-00689-f001:**
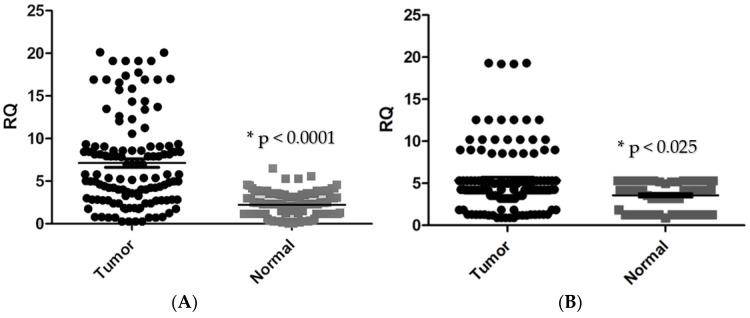
(**A**) Comparison of *CTGF* mRNA expression in tumour and normal tissues (* *p* < 0.0001). (**B**) *MMP-1* mRNA expression is compared in tumour and normal tissues. (* *p* < 0.025). * RQ related to relative quantification.

**Figure 2 genes-13-00689-f002:**
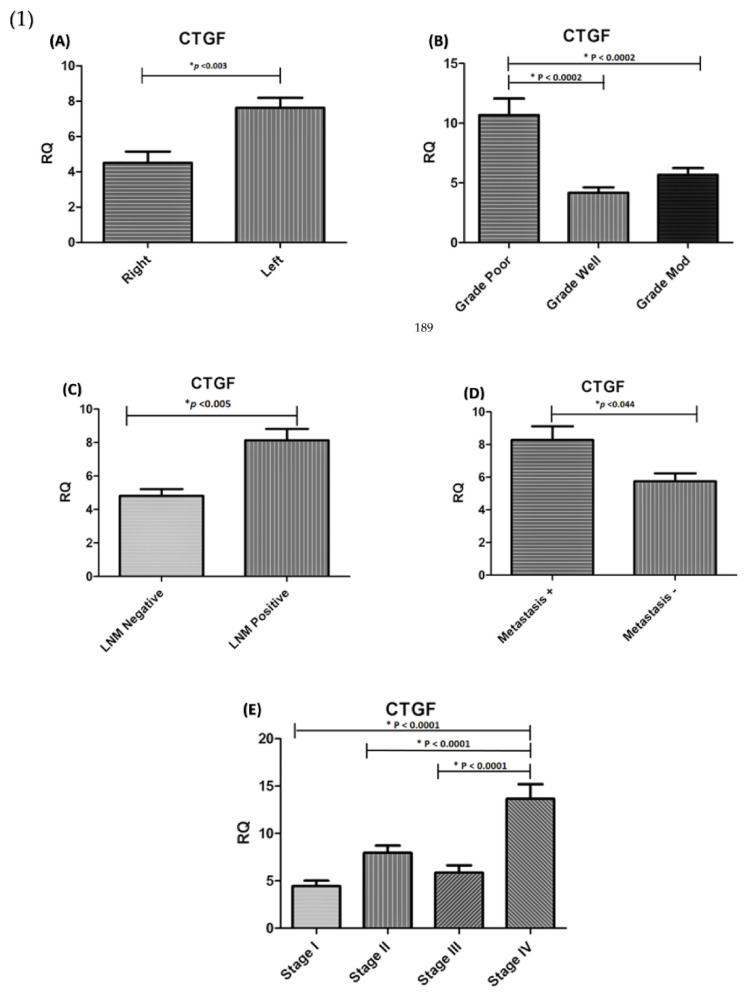
(1) Expression association between CTGF and clinicopathological parameters in CRC. Significant differences were seen in (**A**) Tumour location (*p* < 0.003); (**B**) Tumour grading (*p* < 0.0002); (**C**) lymph node metastasis (*p* < 0.005) and (**D**) distant metastasis (*p* < 0.044); (**E**) Tumour stage (*p* < 0.001).). (2) Association between MMP-1 expression level and clinicopathological parameters in CRC patients. Significant correlation was observed between MMP-1 expression and (**F**) Tumour size (*p* < 0.015) (**G**) Tumour differentiation (*p* < 0.022), (**H**) Tumour stage (*p* < 0.001), (**I**) lymph node metastasis (*p* < 0.028), and (**J**) Metastasis (*p* < 0.012). * RQ related to Relative quantification.

**Figure 3 genes-13-00689-f003:**
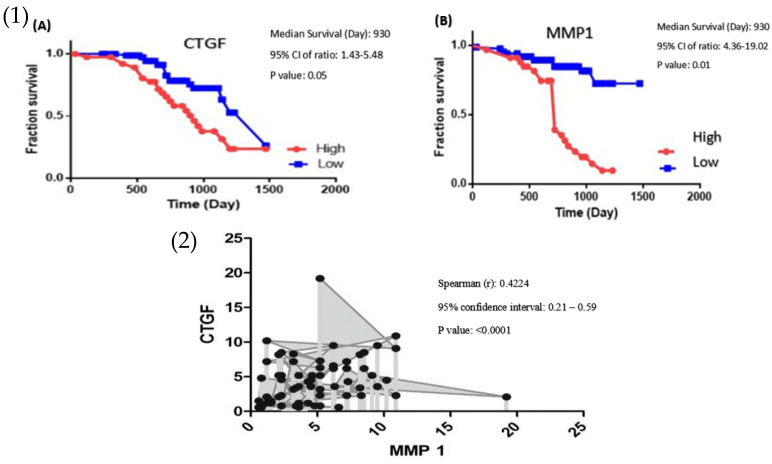
(1) Log Rank Test demonstrate the association between survival rate and the expression level of (**A**) CTGF and (**B**) MMP-1 in patients with colon cancer. (2) Association between the CTGF and MMP-1 expression level (Spearman’s rank test, r = 0.4224, *p* < 0.0001, *n* = 81). r = Spearman’s rank, *n* = Number of XY Pairs.

**Figure 4 genes-13-00689-f004:**
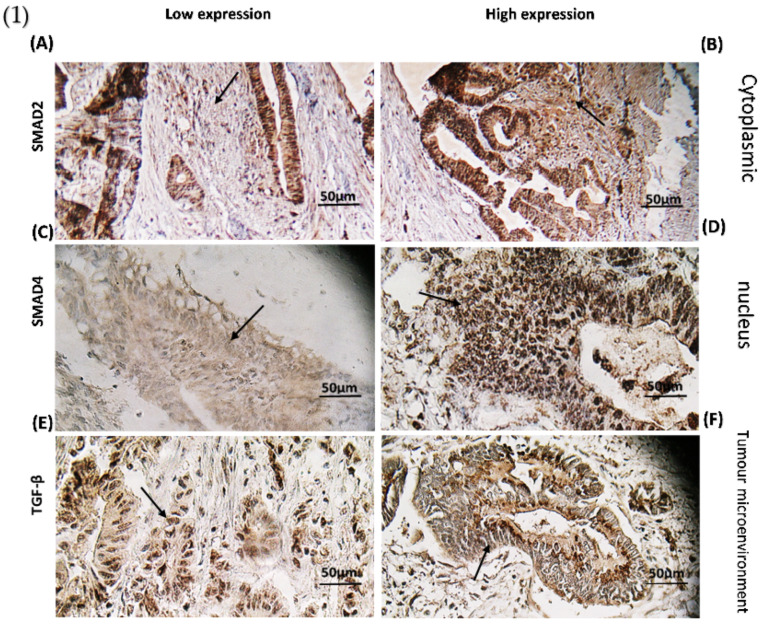
(1) Immunohistochemical staining (**A**) Smad2 cytoplasmic staining low expression; (**B**) Smad2 high expression specimen; (**C**) Smad4 nucleus staining low expression; (**D**) Smad4 high expression specimen; (**E**) IHC staining of TGF-β in Low expressed sample; and (**F**) High TGF-β expression. (2) Association between (**G**) CTGF expression level (* *p* < 0.0002) and (**H**) MMP-1 (* *p* < 0.035) through TGF-β pathway. * RQ related to Relative quantification.

**Table 1 genes-13-00689-t001:** Clinicopathological data of the CRC included patients.

Characteristics	No. of Cases	%
Gender:		
Male	77	(63.1)
Female	45	(36.9)
Age (years):		
≤60 years	55	(45.1)
>60 years	67	(54.9)
Histological grade:		
Grade I (well differentiated)	35	(28.67)
Grade II (moderately differentiated)	70	(57.40)
Grade III (poorly differentiated)	17	(13.93)
Pathological stage:		
Stage I	23	(18.9)
Stage II	47	(38.5)
Stage III	41	(33.6)
Stage VI	11	(9.0)
Lymph node involvement:		
No	48	(39.3)
Yes	74	(60.7)
Tumor size (cm)		
≤3 cm	93	(76.2)
>3 cm	29	(23.8)
Localization:		
Right colon	31	(25.4)
Left colon	91	(74.6)
Status of patient:		
Alive	83	(68.0)
Dead	39	(32.0)
Chemotherapy (After Surgery)		
Yes	97	(79.5)
No	25	(20.5)

**Table 2 genes-13-00689-t002:** Real-time primer sequences.

Gene ID	Primer	Sequence
1490	CTGF	5′-CTGGAAGAGAACATTAAGAAGGGC-3′ 5′-CGGTATGTCTTCATGCTGGTGC-3′
4312	MMP1	5′-GGGAATAAGTACTGGGCTGTTC-3′ 5′-GTCCTTGGGGTATCCGTGTAG-3′
567	β2-micro globulin(B2M)	5′-TGCTGTCTCCATGTTTAGTGTATCT-3′ 5′-TCTCTGCTCCCCACCTCTAAGT-3′

**Table 3 genes-13-00689-t003:** Relationship between the expression of TGF-β signalling pathway and clinicopathological features.

Clinicopathological Features		TGF-β Signalling Pathway (%)		*p*-Value
		Low	High	
Gender:	Male	51 (60.7)	26 (68.4)	0.544
	Female	33 (39.3)	45 (36.9)	
Age	≤60 years	36 (42.9)	19 (50.0)	0.544
	>60 years	48 (57.1)	19 (50.0)	
Tumour size	≤3 cm	65 (77.4)	28 (73.3)	0.653
	>3 cm	19 (22.6)	10 (26.3)	
Location	Right colon	22 (26.2)	9 (23.7)	0.826
	Left colon	62 (73.8)	29 (76.3)	
Differentiation:	Well	28 (33.3)	12 (31.6)	0.847
	Moderate	47 (56.0)	23 (60.5)	
	Poor	9 (10.7)	3 (2.5)	
Pathological stage:	Stage I	19 (22.6)	4 (10.5)	<0.001 *
	Stage II	41 (48.8)	6 (15.8)	
	Stage III	19 (22.6)	22 (57.9)	
	Stage VI	5 (6.0)	6 (15.8)	
Lymph node metastasis	No	23 (27.4)	25 (65.8)	<0.001 *
	Yes	61 (72.6)	13 (34.2)	
Metastasis	Yes	19 (22.6)	33 (86.8)	<0.001 *
	No	65 (77.4)	5 (13.2)	
Family History	Yes	29 (34.5)	12 (31.6)	0.837
	No	55 (65.5)	26 (68.4)	
Recurrence	Yes	11 (13.1)	17 (44.7)	<0.001 *
	No	73 (86.9)	21 (55.3)	

* *p*-value under 0.05 considered significant.

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
