# Peer review of "Oncogenic Role of Connective Tissue Growth Factor Is Associated with Canonical TGF-β Cascade in Colorectal Cancer"

_genes, 2022, doi:10.3390/genes13040689_

Round 1
Reviewer 1 Report
Hosseini et al have analyzed tumor and adjacent normal tissues using real PCR and IHC analyses and find a significant association between TGFB pathway and various attributes of colorectal cancer progression and metastasis. Although TGFB has been extensively studies in CRCs, this manuscript can bring some new insights especially with respect to CTGF.
However, there are several important (major and minor) concerns that I have listed below.
- What is the unit for tumor size? 3 mm? cm?
- How far was the normal tissue from the tumor part? This information is critical.
- In Figure 1, why is there two stars? I would recommend not to write asterisk behind p and instead use *, ** or *** as the regular convention (based on p values)
- In text, CTGF data is discussed first, but in Figure 2 MMP1 is shown first. Fix it.
- In Figure 2, when you have multiple bars in the graph, which bar does the p-value signify? For example, among 4 stages, which stage was significant?
- Expand what RQ stands for somewhere (I assume relative quantitation?).
- Text (line 189) talks about data for correlation between location and MMP1 expression. Where is the bar graph for that (although non-significant)? Or mention ‘data not shown’.
- Again in Figure 3, text mentions CTGF first, but in figure, MMP1 is shown first.
- Explain how to interpret the Figure 3-(2) Spearman’s rank. Is the line running through each points necessary?
- I have some significant concerns about the IHC images in Figure 4-(1).
- The images are not sharp enough
- Why the magnification is different in each?
- Provide a scale bar
- Label the images
- The text mentions the staining is for Smad2 and 4, but which panel represents them? It is extremely confusing to interpret how the results have been written now, and it needs to be re-written. In the figure, clearly label normal, tumor, smad2 or 4, TGFB etc.
- What are the discrepancies mentioned in line 214?
- What do the authors mean by the phrase “in some other kinds of tumors”? (line 221)
- Where is the data for the lines 227-228?
- Again, text mentions CTGF first but in (2) MMP is shown first.
- Few more minor concerns
- Expand CRC at first mention (line 25)
- High and low not aligned in Figure 3 (1).
- What are those blue lines under small text in Figure 3?
- R should not be capital in line 206
- In line 226 where does the parenthesis start?
Author Response
Manuscript ID: Genes-1622364
Title of paper:
Oncogenic Role of Connective Tissue Growth Factor is associated with Canonical TGF-β Cascade in Colorectal Cancer
Thank you very much for sending us the reviewer comments and by including these suggestions we think that the quality of our manuscript will improve to a large extent. We have gone through the reviewer/editorial comments one by one and have included all these comments/suggestions in the revised manuscript. The changes/modifications track changed in the main text.
We hope the revised manuscript will better suit the valuable Genes journal, but we will be happy to consider further revisions. If there are any other queries, please do not hesitate to contact me as a corresponding author.
Sincerely,
Dr. Maziar Ashrafian Bonab
Kent and Medway Medical School, Pears Building, Park Wood Road, Canterbury, Kent, CT2 7FS. ma-ziar.bonab@kmms.ac.uk
#Reviewer1:
- What is the unit for tumor size? 3 mm? cm?
- How far was the normal tissue from the tumor part? This information is critical.
- In Figure 1, why are there two stars? I would recommend not to write asterisk behind p and instead use *, **, or *** as the regular convention (based on p values)
- In-text, CTGF data is discussed first, but in Figure 2 MMP1 is shown first. Fix it.
- In Figure 2, when you have multiple bars in the graph, which bar does the p-value signify? For example, among the 4 stages, which stage was significant?
- Expand what RQ stands for somewhere (I assume relative quantitation?).
- Text (line 189) talks about data for correlation between location and MMP1 expression. Where is the bar graph for that (although non-significant)? Or mention ‘data not shown.
- Again in Figure 3, the text mentions CTGF first, but in the figure, MMP1 is shown first.
- Explain how to interpret Figure 3-(2) Spearman’s rank. Is the line running through each point necessary?
- I have some significant concerns about the IHC images in Figure 4-(1).
- The images are not sharp enough
- Why the magnification is different in each?
- Provide a scale bar
- Label the images
- The text mentions the staining is for Smad2 and 4, but which panel represents them? It is extremely confusing to interpret how the results have been written now, and it needs to be re-written. In the figure, clearly label normal, tumor, smad2 or 4, TGFB, etc.
- What are the discrepancies mentioned in line 214?
- What do the authors mean by the phrase “in some other kinds of tumors”? (line 221)
- Where is the data for lines 227-228?
- Again, the text mentions CTGF first but in (2) MMP is shown first.
- Few more minor concerns
- Expand CRC at first mention (line 25)
- High and low are not aligned in Figure 3 (1).
- What are those blue lines under small text in Figure 3?
- R should not be capital in line 206
- In line 226 where does the parenthesis start?
Q1: What is the unit for tumor size? 3 mm? cm?
R1: The Tumor size proved to be an independent prognostic parameter for patients with colorectal cancer. Optimal cut-off values vary among different parts of the large bowel. The presented tumor size in this article is in cm. [Page 3- Table 1 & Page 5- Table 3]
Q2: How far was the normal tissue from the tumor part? This information is critical.
R2: Normally it is confirmed among all colonoscopists and pathologists that the normal tissue must be removed from 10 cm far apart from tumor tissue. In the present study, all ADJ normal tissues are removed from 10 cm away from tumour region and confirmed by two senior pathologists. [Page 2 – Line 91]
Q3: In Figure 1, why are there two stars? I would recommend not to write asterisk behind p and instead use *, **, or *** as the regular convention (based on p values)
R3: Figure 1 is edited and the significant p-value (p <0.05) in both (A) and (B) blocks is shown as the regular convention. *p-value [Page 6- Figure 1]
Q4: In the text, CTGF data is discussed first, but in Figure 2 MMP1 is shown first. Fix it.
R4: Perfectly correct, there is a mistake when arranging the images. The CTGF plots are settled first as desired according to the discussion in the body of the manuscript and the legend is revised as well. [Page 7,9- Figure 2]
Q5: In Figure 2, when you have multiple bars in the graph, which bar does the p-value signify? For example, among the 4 stages, which stage was significant?
R5: The arrows settled in the graphs show the exact p-value for each bar. [Page 7,9- Figure 2]
Q6: Expand what RQ stands for somewhere (I assume relative quantitation)
R6: RQ means Relative Quantification, which relates the PCR signal of the target transcript in a tumor group to that of another sample such as a normal group. The 2(-delta Ct) method is a convenient way to analyze the relative changes in gene expression from real-time quantitative PCR experiments. In any of the graphs with RQ usage, the description is added under the legend. [Page 6- Figure 1, Page 7,9- Figure 2, Page 16- Figure 4]
Q7: Text (line 189) talks about data for correlation between location and MMP1 expression. Where is the bar graph for that (although non-significant)? Or mention ‘data not shown’
R7: Since the relationship was not significant, a bar graph is not shown in the manuscript, for clarification, ‘data not shown’ is mentioned in the text [Page 12- Line 229]
Q8: Again in Figure 3, the text mentions CTGF first, but in the figure, MMP1 is shown first.
R8: Figure 3 is edited and the CTGF is the first graph as mentioned in the text. [Page 13- Figure 3]
Q9: Explain how to interpret Figure 3-(2) Spearman’s rank. Is the line running through each point necessary?
R9: A Spearman correlation coefficient is also referred to as Spearman rank correlation or Spearman’s rho. It is typically denoted either with the Greek letter rho (ρ), or rs. Like all correlation coefficients, Spearman’s rho measures the strength of association between two variables. As such, the Spearman correlation coefficient is similar to the Pearson correlation coefficient.
All bivariate correlation analyses express the strength of association between two variables in a single value between -1 and +1. This value is called the correlation coefficient. A positive correlation coefficient indicates a positive relationship between the two variables (as values of one variable increase, values of the other variable also increase) while a negative correlation coefficient expresses a negative relationship (as values of one variable increase, values of the other variable decrease). A correlation coefficient of zero indicates that no relationship exists between the variables. However, correlation coefficients like Spearman and Pearson assume a linear relationship between variables. Even if the correlation coefficient is zero, a non-linear relationship might exist.
Compared to the Pearson correlation coefficient, the Spearman correlation does not require continuous-level data (interval or ratio), because it uses ranks instead of assumptions about the distributions of the two variables. This allows us to analyze the association between variables of ordinal measurement levels. Moreover, the Spearman correlation does not assume that the variables are normally distributed. A Spearman correlation analysis can therefore be used in many cases in which the assumptions of the Pearson correlation (continuous-level variables, linearity, heteroscedasticity, and normality) are not met.
Spearman's nonparametric correlation makes no assumption about the distribution of the values, as the calculations are based on ranks, not the actual values.
If you start with a data table with three or more Y columns, you can ask Prism to compute the correlation of each column with each other column, and thus generate a correlation matrix.
The results appear on three pages:
- The correlation coefficient r (or rs). This is computed for each pair of variables and doesn't account for other variables. Prism does not compute a partial correlation coefficient.
- The P-value (two-tail) tests the null hypothesis that the true population correlation coefficient for that pair of variables is zero.
- The number of XY pairs. This might not be the same for all pairs of variables if some data are missing.
Hence in the present statistical analysis, the first page is selected to be presented in the manuscript, it can be changed if necessary.
Q10: I have some significant concerns about the IHC images in Figure 4-(1).
- The images are not sharp enough
The images are changed and become sharper with higher resolution.
- Why the magnification is different in each?
The images are synchronized and all images are used with ×40 magnification.
- Provide a scale bar
Scale bars is missed in the images, now all images are presented with a scale bar
- Label the images
The images are labeled with antibodies used for protein staining, the staining position, and how it is expressed (low & high).
- The text mentions the staining is for Smad2 and 4, but which panel represents them? It is extremely confusing to interpret how the results have been written now, and it needs to be re-written. In the figure, clearly label normal, tumor, smad2 or 4, TGFB, etc.
They are all labeled accordingly.
- What are the discrepancies mentioned in line 214?
The term discrepancies are used in the text for explaining how Smad2 protein is expressed in tissues. In this regard, I decided to change the discrepancies term for clarification of the text. Figure 4 illustrates the different expressions of Smad2,4 and TGFB in tissues by use of the IHC technique. [Page 14 –section 3.4]
Based on our data, 66.40% (81 cases) of tumours had high expression of Smad2 followed by positive TGF-β signalling (described above and 33.60% (41 cases) of tumours represented a low expression of Smad2 with negative TGF-β signalling through cytoplasmic staining. All positive samples had a homogeneous pattern in staining. Figure 4 (A, B) illustrates the low and high expression of Smad2 in the cytoplasm.
- What do the authors mean by the phrase “in some other kinds of tumors”? (line 221)
The terms are corrected for clarification in the text. [Page 14- section 3.4]
- Where is the data for lines 227-228?
Because of not significant data, they are not shown in the figures, hence the phrase “data was not shown” is added to the main text. Although the released data is available can be uploaded as supplementary. [Page 14- Line 279]
- Again, the text mentions CTGF first but in (2) MMP is shown first
Figure 4- (2) part B is corrected and CTGF is presented first.
Q11: Few more minor concerns
- Expand CRC at first mention (line 25)
The CRC abbreviation is expanded. [Page 1- Line 25,26]
- High and low are not aligned in Figure 3 (1).
Figure 3 displays the survival rate according to the expression level of CTGF (A) and MMP-1 (B). In this regard with elevation in the expression level of genes, the survival rates are reduced, overexpression of MMP-1 demonstrates a significant p-value based on upper expression and lower survival.
- What are those blue lines under small text in Figure 3?
Figure 3 is edited completely, and the blue lines are removed. [Page 13- Figure 3]
- R should not be capital in line 206
Thank you for your attention, “TGF-β signalling pathway has a multifaceted role in colorectal cancer” [Page 14- Line 252]
- In line 226 where does the parenthesis start?
The parenthesis is not needed, and it is omitted in the text [Page 14- Line 276]

Reviewer 2 Report
In the present study the Authors aimed to investigate the association between CTGF and MMP-1 mRNA expressions with clinicopathological status and survival rate in CRC patients. Expression levels of MMP-1and CTGF were higher in tumours compared with adjacent normal tissues. Overexpression levels of MMP-1 and CTGF were associated with the lymph node metastasis, distant metastasis, tumour histopathological grading, advanced stage, and poor survival. In addition, a significant association between the upregulation of MMP-1 and tumour location was noted. Upregulation of Smad2 and Smad4 proteins were also significantly correlated with lymph node metastasis, distant metastasis, advanced stage, and poor survival. The results of the study suggest that, canonical TGF-ß signalling regulates both CTGF and MMP-1 expression and CRC progression. Moreover, TGF-ß signalling and its downstream genes could be used as novel biomarkers and novel approaches for targeted therapy in CRC.
Generally speaking, the manuscript is well designed and discussed. Although the paper presents important potential bias regarding the lack of molecular profile information of the analyzed cases (K-RAS status and B-RAF) and mainly the small sample size of casuistry representing the major factor of strength. In my opinion, also in the conclusion section is mandatory introduce a sentence about the need to validate these results by prospective study.
Author Response
Manuscript ID: Genes-1622364
Title of paper:
Oncogenic Role of Connective Tissue Growth Factor is associated with Canonical TGF-β Cascade in Colorectal Cancer
Thank you very much for sending us the reviewer comments and by including these suggestions we think that the quality of our manuscript will improve to a large extent. We have gone through the reviewer/editorial comments one by one and have included all these comments/suggestions in the revised manuscript. The changes/modifications track changed in the main text.
We hope the revised manuscript will better suit the valuable Genes journal, but we will be happy to consider further revisions. If there are any other queries, please do not hesitate to contact me as a corresponding author.
Sincerely,
Dr. Maziar Ashrafian Bonab
Kent and Medway Medical School, Pears Building, Park Wood Road, Canterbury, Kent, CT2 7FS. ma-ziar.bonab@kmms.ac.uk
#Reviewer2:
In the present study, the authors aimed to investigate the association between CTGF and MMP-1 mRNA expressions with clinicopathological status and survival rate in CRC patients. Expression levels of MMP-1and CTGF were higher in tumours compared with adjacent normal tissues. Overexpression levels of MMP-1 and CTGF were associated with lymph node metastasis, distant metastasis, tumour histopathological grading, advanced stage, and poor survival. In addition, a significant association between the upregulation of MMP-1 and tumour location was noted. Upregulation of Smad2 and Smad4 proteins were also significantly correlated with lymph node metastasis, distant metastasis, advanced stage, and poor survival. The results of the study suggest that canonical TGF-ß signalling regulates both CTGF and MMP-1 expression and CRC progression. Moreover, TGF-ß signalling and its downstream genes could be used as novel biomarkers and novel approaches for targeted therapy in CRC.
Generally speaking, the manuscript is well designed and discussed. Although the paper presents important potential bias regarding the lack of molecular profile information of the analyzed cases (K-RAS status and B-RAF) and mainly the small sample size of casuistry representing the major factor of strength. In my opinion, also in the conclusion section is mandatory to introduce a sentence about the need to validate these results by a prospective study.
The aim of the present study is to focus especially on the TGF-ß signalling pathway, particularly on canonical SMAD dependent cascade. Otherwise, there are more different signalling pathways that are occasionally involved in CRC development and progression. This is the main cause of not presenting any data regarding (K-RAS status and B-RAF). Although many of the research articles have some limitations in the studies, analysing and presenting the molecular profile information of (K-RAS status and B-RAF) participated cases in this study is our principles that limit the extent of our data. We will discuss the concerns in upcoming proposals for improving the next results with prospective studies.
According to the calculated sample size in this study, statistics specialists used the cross-sectional formula based on the Iranian population. The formula is provided below. The sample size is calculated at 100. In the present study, we used 122 FFPE samples.
z= The normal value of the standard unit, which at 95% confidence level is 1.96
p= The prevalence of colorectal cancer
c=precision corresponding to effect size
There are different research articles based on the TGF-ß signalling pathway in CRC patients which are smaller sample sizes with 50 and 80 samples.
- https://doi.org/10.1111/cas.14444
- https://doi.org/10.3892/ijo.2018.4591

Round 2
Reviewer 1 Report
The authors have satisfactorily addressed all of my concerns. However, there is a need for thorough check for English grammar and syntax. I would recommend them to get the manuscript checked from a native English speaker, or from a professional firm. There are still many instances, I have listed couple of them below:
"*RQ related to Relative Quantification" can be changed to
RQ: Relative quantification
&
"data was not shown" to data not shown
etc.
Author Response
Dear Sir/Madam
Thank you for the second review comments on our manuscript entitled " Oncogenic Role of Connective Tissue Growth Factor is associated with Canonical TGF-β Cascade in Colorectal Cancer" (Manuscript ID:1622364). The comments are very helpful for revising and improving our paper, as well as the important guidance significance to other researchers. We have studied the comments carefully and made corrections which we hope meet with approval.
The first reviewer was concerned about the need for a manuscript thorough check for English grammar and syntax and recommendation to get the manuscript checked by a native English speaker, or from a professional firm, I confirm that the whole paper is proof-read 3 times by our native English speaker colleagues for resolving any grammatical points left before sending the last version of the manuscript for the Genes journal.
Thank you for your help and consideration in advance.
Dr Maziar Ashrafian Bonab
Senior Lecturer/ ML Scientific Basis of Medicine
T: +44-07732560007
E: maziar.bonab@kmms.ac.uk
www.kmms.ac.uk (Canterbury Christ Church University and the University of Kent in collaboration)
▪ Canterbury Christ Church University, Augustine House AH1.26, Rhodaus Town, Canterbury, Kent CT1 2YA
▪ University of Kent, Pears Building, Park Wood Road, Canterbury, Kent, CT2 7FS

Reviewer 2 Report
The manuscript is characterized by a reasonable technical quality and the presentation is acceptable, but it does not present any new solid information. Due to the lack of KRAS and B-RAF status, the potential selection bias within these series I am not convinced that MMP-1 and CTGF have any impact on the course of the disease. This report, in this form, does not advance our knowledge by any means, in any form. For that reason, my enthusiasm for the manuscript is significantly dampened
Author Response
Dear Sir/Madam
Thank you for the second review comments on our manuscript entitled " Oncogenic Role of Connective Tissue Growth Factor is associated with Canonical TGF-β Cascade in Colorectal Cancer" (Manuscript ID:1622364). The comments are very helpful for revising and improving our paper, as well as the important guiding significance to other researchers. We have studied the comments carefully and made corrections which we hope meet with approval.
The second reviewer is concerned about not presenting any new solid information, due to the lack of KRAS and B-RAF status, a brief detail about TGF-β signaling is presented below:
TGF-β signaling is complex and mediates both pro- and anti-tumoural activities in cancer cells depending on their context, in space and time, and their microenvironment. Over the last decade, research has increasingly focused on the microenvironment surrounding cancer cells, and their role in tumour development and progression.
TGF-β is a well-recognized factor of development and is involved in the regulation of cell proliferation, differentiation, invasion, and inflammation. Hijacking crucial biological functions by deregulating the TGF-β signaling pathway has recently emerged as a leading area of preclinical and clinical cancer research. On one hand, the TGF-β pathway promotes cell cycle arrest, apoptosis, and autophagy in epithelial cells, and also inhibits inflammation. On the other hand, by promoting angiogenesis, cell motility, invasion, EMT, or cell stemness, the TGF-β pathway promotes tumour progression.
All of the ligands of the TGFβ family are initially synthesized and secreted as latent precursors that need to be proteolytically processed by extra-cellular pro-protein convertases to produce biologically active dimeric ligands. The signal transduction mechanism for all TGF is similar. It requires ligand binding to one of the five distinct members of constitutively activated membranous type II serine/threonine kinase receptors, and subsequent recruitment and transphosphorylation of one of the seven types I serine/threonine kinase receptors, also known as activin receptor-like kinases (ALK1-7) (Wakefield & Hill, 2013). For some ligands, additional co-receptors are required for optimal ligand binding and activation of the type I-type II receptor heterodimer. After phosphorylation, the type I receptor kinase domain becomes activated and can then activate the canonical (SMAD-dependent) pathway through phosphorylation of the receptor-regulated SMAD proteins (R-SMAD). It is classically admitted that TGFβs, activins, and NODAL signal through SMAD2 andSMAD3, and BMPs and GDFs through SMAD1, SMAD5, and SMAD8, but TGFβs can also induce phosphorylation of SMAD1 and SMAD5 (Wakefield & Hill, 2013). The R-SMADs form heteromeric complexes with SMAD4 in early endosomes, through a clathrin-dependent internalization pathway that requires accessory proteins such as SARA (Bierie & Moses, 2006; Ikushima & Miyazono, 2010). These complexes then translocate and accumulate into the nucleus and bind to site-specific recognition sequences within the promoter regions of hundreds of target genes to directly regulate their transcription, both positively and negatively. Various other DNA binding co-factors such as p300, CBP, and FOXH1 can interact with SMAD complexes to amplify SMAD-dependent gene transcription. The TGFβ superfamily pathway activities are subject to numerous levels of regulation: interaction of the ligands with extracellular antagonists that prevent their binding to receptors; modulation of R-SMAD stability through phosphorylation by MAPKs, glycogen synthase kinase 3β (GSK3β) and cyclin-dependent kinases (CDKs); Smurf (SMAD-ubiquitination-regulatory factor)-dependent degradation; or expression of inhibitory SMAD proteins (I-SMAD), SMAD6 and SMAD7 (Wakefield & Hill, 2013).In the non-canonical pathway, TGFβ signaling activates SMAD-independent pathways such as PI3K/AKT, MAPK pathways (ERK, JNK, and p38 MAPK), c-Src, NF-κB, FAK, Abl, or small GTPases such as RhoA, Rac1 and Cdc42. Moreover, transversal signaling, especially at the SMAD level, allows TGFβ pathway activation to integrate signals from integrins, Wnt/β-catenin, Notch, Hedgehog, TNF-α, or EGF-dependent pathways (Watabe & Miyazono, 2009; Sakaki-Yumoto et al., 2013). In addition, CTGF is a transcriptional target of TGF-β signalling and induces angiogenesis and invasion by the regulation of matrix metalloproteinases (MMPs), which are extracellular matrix (ECM) degradation enzymes and play key roles in tumour progression through digestion of ECM and providing a suitable condition for tumour angiogenesis, invasion, and metastasis.
Summary: In the canonical signaling pathway, biologically active TGF-β ligands bind to TGFβRII, which in turn activates TGFβRI.TGFβRI-regulated SMAD2/3 proteins are phosphorylated at their C-terminal serine residues and form complexes with SMAD4 (co-SMAD), initiating a number of biological processes through transcriptional regulation of target genes. (B) In the non-canonical signaling pathways, the TGF-βreceptor complex transmits its signal through other factors, such as the mitogen-activated protein kinases (MAPKs), phosphatidylinositide 3-kinase (PI3K), TNF receptor-associated factor 4/6 (TRAF4/6) and Rho family of small GTPases. Activated MAPKs can exert transcriptional regulation either through direct interaction with the nuclear SMAD protein complex or via other downstream proteins.
The first reason for NOT presenting any data about KRAS and B-RAF status: By all presented detail, we focus specifically on Smad dependent or canonical TGF-β signaling pathway.
The literature review was performed. There are newly published articles with specific orientations about canonical or non-canonical TGF-β signaling pathways. Some papers are presented below:
https://doi.org/10.1158/1078-0432.CCR-09-3256
Published online 2022 Feb 7. doi: 10.3748/wjg.v28.i5.547
Published online 2020 Jan 16. doi: 10.7150/thno.39740
The second reason for NOT presenting any data about KRAS and B-RAF status: In another hand, the molecular data for half of the participated patients is accessible, all entered in the prism software and analyzed based on the technical method. No significant data was released based on a lack of sufficient available data or many other reasons. Hence, all the authors become consistence in presenting the available TGF-β Smad canonical signaling data and not the incomplete non-canonical pathway. We hope in our next design proposal both canonical and non-canonical TGF-β signaling pathways investigate thoroughly.
Please see the relevant images in the attached document.
Thank you for your help and consideration in advance.
Dr Maziar Ashrafian Bonab
Senior Lecturer/ ML Scientific Basis of Medicine
T: +44-07732560007
E: maziar.bonab@kmms.ac.uk
www.kmms.ac.uk (Canterbury Christ Church University and the University of Kent in collaboration)
▪ Canterbury Christ Church University, Augustine House AH1.26, Rhodaus Town, Canterbury, Kent CT1 2YA
▪ University of Kent, Pears Building, Park Wood Road, Canterbury, Kent, CT2 7FS
